# Effects of Soybean Isoflavone and Astragalus Polysaccharide Mixture on Colostrum Components, Serum Antioxidant, Immune and Hormone Levels of Lactating Sows

**DOI:** 10.3390/ani11010132

**Published:** 2021-01-08

**Authors:** Hongzhi Wu, Ji Yang, Sibo Wang, Xin Zhang, Jinwang Hou, Fei Xu, Zhilong Wang, Li Xu, Xinping Diao

**Affiliations:** College of Animal Science and Technology, Northeast Agricultural University, Harbin 150030, China; hong-zhi@163.com (H.W.); yangji209@163.com (J.Y.); wang1014vip@163.com (S.W.); zhx76098@163.com (X.Z.); hjw970214@163.com (J.H.); fei_xu2020@163.com (F.X.); w188zl188@163.com (Z.W.)

**Keywords:** lactating sows, soybean isoflavones, astragalus polysaccharides, colostrum components, immune and hormone level

## Abstract

**Simple Summary:**

The reproductive performance of sows plays an important role in the development of the pig industry. The health status during gestation and lactation capacity of sows determines the healthy development and later growth status of piglets. This study aimed to investigate the effects of soybean isoflavone and astragalus polysaccharide mixture on the colostrum components, and serum antioxidant activities, immune function, and hormone levels of lactating sows. Results showed that the soybean isoflavone and astragalus polysaccharide mixture increased the total lactation yield in the lactating phase, and improved the antioxidant, immune and hormone levels in the serum of lactating sows. In conclusion, the soybean isoflavone and astragalus polysaccharide mixture can improve the average daily feed intake, lactation yield, serum antioxidant activity, immune function, and the hormone levels of lactating sows, and the optimum dosage in this study was 200 mg/kg. This study provides a theoretical basis for the application of the soybean isoflavone and astragalus polysaccharide mixture application in the production of lactating sows and a reference basis for the development of green feed additives.

**Abstract:**

The objectives of this study were to investigate the effects of soybean isoflavone (SI) and astragalus polysaccharide (APS) mixture on the colostrum components, serum antioxidant, immune and hormone levels of lactating sows. A total of 72 healthy Yorkshire × Landrace lactating sows, were randomly divided into four treatments with six replicates and three lactating sows for each replicate. The control group was fed the basal diet, while the experimental groups were fed the basal diet with 100, 200 and 300 mg/kg SI and APS mixture in the form of powder, respectively. Compared with the control group, (a) the total lactation yield of the 200 mg/kg group was significantly higher (*p* < 0.05) at 21 days, (b) there was no significant difference in colostrum composition, (c) TG, CHO and MDA content in each treatment group were significantly decreased (*p* < 0.05), (d) IgA, GH, IGF-1, TNF-α and SOD contents in the 200 mg/kg group were significantly increased (*p* < 0.05). The SI and APS mixture could improve the average daily feed intake, lactation yield, serum antioxidant activities, immune function, and hormone levels of lactating sows, and the optimum dosage in this study was 200 mg/kg.

## 1. Introduction

During late gestation and the entire lactation period, the nutrition metabolism and physiological state of sows will change to satisfy the normal growth of the fetus in the womb and piglets, which will reinforce maternal oxidative stress, leading to insufficient lactation, anorexia, frail and declining immunity of sows, causing hidden troubles for the late reproduction of sows [1,2,3,4]. Porcine milk is the main source of nutrition for piglets during lactation, especially the colostrum, which is a crucial material for piglets to gain passive immunoglobulins and improve immune function after piglets are exposed to environmental cold and pathogens [5,6,7,8]. Therefore, increasing porcine milk yield and quality is particularly important to keep the best production performance of sows. Soy Isoflavones (SI), a natural active phytoestrogen, mainly found in legumes in nature, can affect sow hormones, metabolic biological activity, protein synthesis, and growth factor activity [9,10]. Astragalus polysaccharides (APS), i.e., a polysaccharide component isolated and extracted from the root of astragalus, is composed of glucose, galactose, and arabinose, among others, and has a strong immune-promoting effect [11,12,13].

This study intends to (1) study the effects of SI and APS mixture on the serum physiological indicators, and colostrum composition, of lactating sows by adding SI and APS mixture to lactating sow diets, (2) discuss the feasibility and appropriate contents of SI and APS mixture in the production of lactating sows, and (3) preliminarily explore the impact mechanism of the SI and APS mixture on production performance of lactating sows, to further develop and use the SI and APS mixture, and accumulate experience and theoretical basis for safe and reliable additives.

## 2. Materials and Methods

This study was approved by the Animal Care and Use Committee of Northeast Agricultural University (NEAU-2011-9). Animals used in these experiments were cared for following the guidelines stated in the Guide for the Care and Use of Agricultural Animals in Agricultural Research and Teaching of Heilongjiang Province in China.

### 2.1. Experimental Material

Gestating and lactating sows: Yorkshire × Landrace, with second parity, same genetic background and similar delivery date.

SI and APS mixture: SI and APS were purchased from Shanxi Haoyang Biotechnology Co., Ltd. SI purity was 40.35% and its chemical substance registration number was 574-12-9. APS purity was 55% and its chemical substance registration number was 9005-38-3. SI: APS ratio in the mixture was 1:5, in the form of powder, prepared according to the appropriate range of SI and APS in sow production, respectively.

Enzyme-linked immunoassay kits were purchased from Shanghai Sangon Biotechnology Co., Ltd (Shanghai, China).

### 2.2. Experiment Design and Sample Collection

In an individual stall, 72 healthy Yorkshire × Landrace lactating sows, were randomly divided into four treatments with six replicates, with three lactating sows in each replicate. The control group (GC) was fed the basal diet, while the experimental groups were fed the basal diet with 100, 200, and 300 mg/kg SI and APS mixture in the form of powder (GC100, GC200, GC300, respectively). The composition (kg/100 kg) of the basal experimental diets for gestating and lactating sows is shown in Table 1. In the farrowing room, each group was fed the corresponding experimental diet from the 107th day of gestation, restricted to 3.0 kg per day for 3 days before the previewed farrowing date, and fed twice a day at 8:00 and 16:00. Every sow was fed 2.0 kg on the first day after farrowing, and increased by 0.5–1 kg per day thereafter. Then, the sows were fed close to ad libitum. Each sow farrowed 10.01 ± 0.16 piglets, and piglets were kept warm with thermal insulation and warming lamps. Both sows and piglets drank freely. The piglet number was adjusted to 10 per lactating sow, with body weight 1.45 ± 0.15 kg, in 3 days after farrowing for cross-fostering, then 1 lactating sow and 10 adjusted piglets were feeding in the individual stall. The other feeding management and immunization procedures, including ventilation of the piggery, zinc supplementation (200 mg zinc lysine particles mixed in 250 kg water for daily drinking), and iron (2 mL iron dextran injection per piglet were injected into each piglet on the 3rd day of lactation) supplementation for pigs, were carried out following the pig farm guidelines.

The experiment lasted from the 107th day of gestation to the weaning of piglets on the 21st day. Feed intake of sows, weight gain, diarrhea ratio (ratio of piglets with diarrhea to the total number of piglets in each group), and mortality ratio of piglets were recorded during the experiment period. The lactation yield of sows was indirectly calculated by the body gain (Equation (1)) of piglets during the experiment.
Lactation yield = (*F_BW_* − *B_BW_*) × 0.004(1)
where *F_BW_* represents the final body weight (kg) of 10 piglets on the 21st day, *B_BW_* represents the birth body weight (kg) of 10 piglets on the farrowing day, 0.004 represents 0.004 kg milk needed for every kilogram body weight gain of piglets.

A total of 10 mL of colostrum was collected from the front, middle and rear nipples of the sows 2 h after the farrowing ended, then mixed well and stored at −20 °C until further testing. On the 1st, 10th, and 21st days of lactation, blood (10 mL) was collected with disposable vacuum blood collection tubes from the ear vein of each sow. After resting the blood for 15 min, it was centrifuged at 3000 rpm for 15 min to obtain the serum, which was divided into Eppendorf tubes and stored at −20 °C for testing.

### 2.3. Colostrum Composition Determination

An infrared milk component analyzer (Milko-Scan 134 A/B, Foss, Denmark) was used to measure the fat, lactose, and protein content.

### 2.4. Serum Index Determination

Total protein kit, No. C006225, was used to detect the total protein (TP), with the coefficients of variation of inter-and intra-assay being 4.98 and 4.66%, respectively.

Albumin kit, No. D711271, was used to detect the albumin (ALB), with the coefficients of variation of inter-and intra-assay being 3.98 and 4.76%, respectively.

Urea nitrogen kit, No. D799849, was used to detect the urea nitrogen (UN), with the coefficients of variation of inter-and intra-assay being 4.48 and 4.96%, respectively.

Glucose kit, No. A501991, was used to detect the glucose (G), with the coefficients of variation of inter-and intra-assay being 4.55 and 4.36%, respectively.

Triglyceride kit, No. D799795, was used to detect the triglyceride (TG), with the coefficients of variation of inter-and intra-assay being 3.55 and 4.26%, respectively.

Total cholesterol kit, No. A506234, was used to detect the total cholesterol (CHO), with the coefficients of variation of inter-and intra-assay being 3.95 and 4.56%, respectively.

Immunoglobulin G kit, No. D711074, was used to detect the immunoglobulin G (IgG), with the coefficients of variation of inter-and intra-assay being 2.55 and 2.26%, respectively.

Immunoglobulin A kit, No. D711189, was used to detect the immunoglobulin A (IgA), with the coefficients of variation of inter-and intra-assay being 3.39 and 3.96%, respectively.

Interleukin-2 kit, No. D721020, was used to detect the interleukin-2 (IL-2), with the coefficients of variation of inter-and intra-assay being 4.26 and 4.33%, respectively.

Tumor Necrosis Factor-α kit, No. C600021, was used to detect the Tumor Necrosis Factor-α (TNF-α), with the coefficients of variation of inter-and intra-assay being 4.46 and 4.39%, respectively.

Complement 4 kit, No. D711072, was used to detect the complement 4 (C4), with the coefficients of variation of inter-and intra-assay being 4.46 and 4.39%, respectively.

Prolactin kit, No. D711066, was used to detect the prolactin (PRL), with the coefficients of variation of inter-and intra-assay being 4.59 and 4.65%, respectively.

Growth hormone kit, No. D711077, was used to detect the growth hormone (GH), with the coefficients of variation of inter-and intra-assay being 4.86 and 4.95%, respectively.

Insulin-like growth factor-1 kit, No. D711043, was used to detect the insulin-like growth factor-1 (IGF-1), with the coefficients of variation of inter-and intra-assay being 4.86 and 4.95%, respectively.

Total antioxidant capacity kit, No. D799275, was used to detect the total antioxidant capacity (T-AOC), with the coefficients of variation of inter-and intra-assay being 4.50 and 4.65%, respectively.

Activities of superoxide dismutase kit, No. D799593, was used to detect the activities of superoxide dismutase (SOD), with the coefficients of variation of inter-and intra-assay being 4.30 and 4.45%, respectively.

Glutathione peroxidase kit, No. D751008, was used to detect the glutathione peroxidase (GSH-Px), with the coefficients of variation of inter-and intra-assay being 4.39 and 4.46%, respectively.

Catalase kit, No. D799579, was used to detect the catalase (CAT), with the coefficients of variation of inter-and intra-assay being 4.10 and 4.26%, respectively.

Malondialdehyde kit, No. D751027, was used to detect the content of malondialdehyde (MDA), with the coefficients of variation of inter-and intra-assay being 4.50 and 4.76%, respectively.

The TP, ALB, UN, G, TG, CHO were measured with a fully automatic biochemical analyzer, CG3040B, Changchun Guangji Medical Instrument Co., Ltd. (Jilin, China). The IgG, IgA, IL-2, TNF-a, C4, PRL, GH, IGF-1, T-AOC, SOD, GSH-Px, CAT, and MDA were determined with the multifunctional marker (SuPerMax 3100, Shanghai, China), enzyme-linked immunoassay kits were purchased from Shanghai Sangon Biotechnology Co., Ltd. (Shanghai, China). The relevant determination operations were carried out in strict accordance with the kit instructions.

### 2.5. Statistical Analysis

Statistical analyses were conducted using SPSS statistics software (version 20.0, International Business Machines Corporation, Armonk, NY, USA). Data were expressed as mean ± SEM. Statistical comparisons of different treatments were performed using one-way ANOVA. The test results of all analyses were considered significant at *p* < 0.05.

## 3. Results

### 3.1. Effects of SI and APS Mixture on Production Performance of Lactating Sows

The average daily feed intake of lactating sows in GC100 and GC200 was significantly higher (*p* < 0.05) than that in GC. The total lactation yield in GC200 was significantly higher (*p* < 0.05) than that in the GC and GC300. There was no significant difference (*p* > 0.05) between groups in the average daily gain, diarrhea ratio, and mortality ratio of piglets (Table 2).

### 3.2. Effects of SI and APS Mixture on Colostrum Composition of Lactating Sows

On the day of delivery, there were no significant differences (*p* > 0.05) in fat, lactose, and protein contents in colostrum between groups (Table 3).

### 3.3. Effects SI and APS Mixture on Serum Antioxidant and Other Indexes of Lactating Sows

On the 1st day of lactation, there were no significant differences (*p* > 0.05) in TP and G contents between groups; ALB contents in the GC200 and GC300 were higher (*p* < 0.05) compared to those in GC. UN contents in treatment groups, especially in GC200, were significantly lower (*p* < 0.05) than those in GC. The TG and CHO contents in groups with SI and APS mixture were significantly lower (*p* < 0.05) compared to GC (Table 4).

On the 10th day of lactation, there were no significant differences (*p* > 0.05) in ALB and G contents between groups; TP content in the GC200 was significantly higher (*p* < 0.05) than that in GC. The UN contents in GC100 and GC200 was significantly lower (*p* < 0.05) than those in GC. The TG and CHO contents in groups with SI and APS mixture were significantly lower (*p* < 0.05) than those in GC (Table 4).

On the 21st day of lactation, there were no significant differences (*p* > 0.05) in ALB and G contents between groups. TP content in GC200 was higher (*p* < 0.05) than that in GC. The UN, TG, and CHO contents in the groups with SI and APS mixture were significantly lower (*p* < 0.05) than those in GC (Table 4).

### 3.4. Effects of SI and APS Mixture on Serum Immune Indexes of Lactating Sows

On the 1st day of lactation, there were no significant differences (*p* > 0.05) in IL-2, C4, and IgG contents between groups. Compared with those in GC, TNF-α contents in GC200 were higher (*p* < 0.05), and IgA contents in GC200 and GC300 were higher (*p* < 0.05) (Table 5).

On the 10th day of lactation, the IL-2 contents in GC100, GC200, and GC300 were higher (*p* < 0.05) compared with those in GC. TNF-α content in GC200 was higher (*p* < 0.05) than that in GC. The C4 content in GC300 was higher (*p* < 0.05) compared with that in GC. IgG and IgA contents in GC200 were higher (*p* < 0.05) than those in GC, and IgA content was also higher in GC200 when compared to the other supplementation groups (Table 5).

On the 21st day of lactation, IL-2 contents in GC200 and GC300 were higher (*p* < 0.05) than those in GC. The TNF-α, IgG, and IgA contents in GC200 were higher (*p* < 0.05) than those in GC, and IgA contents were higher in GC200 when compared to GC100 or GC300. Finally, there were no significant differences (*p* > 0.05) in C4 contents between groups (Table 5).

### 3.5. Effects of SI and APS Mixture on Serum Hormone Level of Lactating Sows

On the 1st day of lactation, the PRL, IGF-1, and GSH-Px contents in GC200 and GC300 were higher (*p* < 0.05) compared with those in GC. The MDA contents in GC200 and GC300 were lower (*p* < 0.05) compared with those in GC. The GH, T-AOC, and SOD contents in GC200 and GC300 were higher (*p* < 0.05) compared with those in GC. There were no significant differences (*p* > 0.05) in CAT contents between groups (Table 6).

On the 10th day of lactation, the PRL, GH, IGF-1, and SOD contents in GC200 and GC300 were higher (*p* < 0.05) compared with those in GC. The MDA content in GC200 and GC300 were lower (*p* < 0.05) compared with that in GC. Furthermore, there were no significant differences (*p* > 0.05) in CAT and T-AOC contents between groups (Table 6).

On the 21st day of lactation, the PRL, GH, IGF-1, T-AOC, SOD, and GSH-Px contents in GC200 and GC300 were higher than those in GC. The CAT content in GC200 was higher (*p* < 0.05) than that in GC. Furthermore, the MDA content in GC200 was lower (*p* < 0.05) than that in GC (Table 6).

## 4. Discussion

### 4.1. Effects of SI and APS Mixture on Production Performance of Lactating Sows

Greiner et al. [14] reported that genistein, a flavone, and daidzein, a soy isoflavone, significantly improved the production performance of piglets. Similarly, evidence suggested that adding glycitein, isoflavone compounds, to the diets of lactating sows enhanced the growth performance of the sucking piglets [15]. Meanwhile, Li [16] found that SI caused no significant improvement in the production performance of suckling pigs. Feeding APS can improve the growth performance of large yellow croaker [17], and Wang [18] found that the APS together with ginseng polysaccharide increased the average daily gain and decrease the feed conversation ratio of piglets. In this study, the SI and APS mixture significantly improved the average daily feed intake and lactation yield of lactation sows. This is inconsistent with Li’s [16] results, which might be a result of the interaction effect of SI and APS. SI is structurally similar to mammalian estrogens and therefore may act as estrogen agonists [19], which can act on the gonadal axis to promote testosterone and prolactin secretion in the lactating sows [20], which improves the lactation yield and can reduce piglet mortality. In Wang’s [18] study, APS and ginseng polysaccharide increased the survival rate and average daily gain of piglets. Franke [21] reported that SI could be detected in the milk of women who ate soybeans, indicating that SI could enter the milk through body metabolism. Our results showed that there were no significant differences in diarrhea and mortality ratio between groups, which is not consistent with Wang’s [18] results. This could be explained by that the piglets’ digestive systems are underdeveloped and normally lack of necessary intestinal microorganisms, which made them unable to fully utilize the SI and APS from milk, and another reason may be that the SI was less effective when compared to ginseng polysaccharide on regulating the physiological function of piglets.

### 4.2. Effects of SI and APS Mixture on Colostrum Composition of Lactating Sows

Porcine milk is a direct source of energy and protein, so the quantity and quality of milk will directly affect the health and growth of piglets [3,5,22]. Porcine milk is produced in the mammary gland epithelial cells by absorbing the corresponding nutrients from the flowing blood [5]. Then, the synthesized milk is secreted into the mammary cells and ducts under the action of the corresponding hormone, including prolactin and progestational hormone, and taken by the piglets [23]. Therefore, the porcine milk composition reflects the nutritional status of lactating sows. Li [24] observed that the fat, lactose, and protein contents in colostrum and regular milk increased as the level of SI in the diet increased. Similarly, Tan [25] showed that feeding APS enriched the composition of sow colostrum. Our study showed that there were no significant differences in fat, lactose, and protein contents in colostrum between groups, which differs from the findings above. The discrepancy in the compositions of colostrum may be attributed to the relatively short period of SI and APS mixture ingestion during gestation in this study, which is 7 days prior to the predicted farrowing date. As a result, there may not be enough time for mammary cells to utilize them.

### 4.3. Effects of SI and APS Mixture on Serum Antioxidant and Other Indexes of Lactating Sows

In the animal body, protein absorption, synthesis, and metabolism could be reflected by the serum TP and ALB content, which plays an important role in maintaining plasma colloidal osmotic pressure, normal plasma values, transportation, catalysis, nutrition, and other functions [26]. ALB is mainly synthesized by the liver. The content of ALB reflects the liver function and the immune system status and has been used to diagnose liver diseases and lesions. The final product of protein metabolism in the animal body is UN, which is the major indicator of animal health status, and has been used to measure the balance of amino acids in the diet [27,28]. In animals, protein metabolic conditions can be reflected by UN content. Increases in serum UN content means enhanced decomposition of the protein and decreased nitrogen deposition. Animals with high serum UN content may be in the disease or the sub-health status. The SI and APS mixture increased the TP and ALB levels and reduced UN content in the plasm of mice, indicating that SI and APS mixture could reduce urine nitrogen excretion, increase protein reserve, provide essential amino acids, and prevent damage to the animal body [16]. Additionally, in the three treatment groups, TP and ALB contents were higher than those in GC, while UN content was significantly lower than that in GC during the experimental period, which was consistent with Zhilong’s study results [29]. Moreover, the SI and APS mixture did not display adverse effects on protein metabolism of lactating sows, possibly because it could also improve the nutritional metabolism of sows and maintain them in a healthy state.

The SI and APS have the function of lowering blood glucose and blood lipid in animals [24,30,31]. During the whole experimental period, the TG and CHO contents in GC100, GC200, and GC300 were significantly lower than those in GC, but there was no significant difference in G content between groups. Lin [32] observed that genistein promoted glucose absorption of isolated small intestinal epithelial cells. Kim [30] noted that SI reduced triglyceride levels in the blood, and Zheng [13] found that APS significantly reduced the TG and CHO contents in the blood, causing no harm to the health state of animals. Our results of TG and CHO contents in blood were consistent with previous reported, while the G content in blood was not consistent with previous findings [24,29]. The inconsistency may result from the interaction of SI and APS, which neutralizes the effects on glucose absorption and transport as the effect of individual substance alone, i.e., either SI or APS, was better than their combination for glucose metabolism [30,31]. Ali [33] proposed that SI lowered plasma CHO and that this hypocholesterolemic effect appeared to be due in part to the modulation of steroid hormones, and SI also promoted lipid transport and excretion, thereby reducing blood glucose and blood lipids. Additionally, APS can increase the transcription and expression of ABCA1 genes in cells, promoting the outflow of body CHO, and reducing the amount of deposition [34].

### 4.4. Effects of SI and APS Mixture on Serum Immune Indexes of Lactating Sows

IL-2, mainly secreted by Th1 T cells, promotes the growth of immune T cells, mediates cellular immunity and inflammatory responses, and regulates the humoral body’s immune response [35]. Results indicated that the SI and APS mixture can increase the IL-2 contents in serum on the 10th and 21st day of lactation, which were consistent with results reported by Chen [36], i.e., SI can upregulate the IL-2 content in the body and those by Wang [37], i.e., APS can increase the IL-2 content in broilers.

TNF-α, an inflammatory cytokine, plays a key role in preventing mastitis and oncogenic processes. Besides, it can cause the accumulation of phagocytic proteins, induce the differentiation of immunotoxic T cells, enhance the toxic and side effects of monocytes, and activate lymphocyte activity [38,39]. During the experimental period, the TNF-α content in GC200 was significantly higher than that in GC, indicating that SI and APS mixture significantly promote the secretion of TNF-α, which can lead to the accumulation of phagocytic cell protein, and activate the activity of lymphokines, reducing the probability of inflammatory reactions such as mastitis in sows. Gaffer [40] reported that SI could increase the TNF-α content to enhance the immunity function of rats. Additionally, Chen [41] found that APS could significantly promote the production of TNF-α in peritoneal macrophages, both results being consistent with those of the present experiment.

Immunoglobulins are a kind of globulin produced by the animal body that is stimulated by antigens. It has biological functions such as binding antigens and activating complement. IgG, one of the main immunoglobulin in serum, involves regulating antibacterial, antiviral, and other immune activities via agglutinating and/or precipitating antigens [42,43]. IgA is the main immunoglobulin in animal exudates. In this study, IgA and IgG contents in GC200 were higher than those in GC, except that the difference was not significant for IgG on the 1st day of lactation, indicating that the SI and APS mixture could improve the immune ability of lactating sows, which was consistent with Zhilong’s [29] report.

The complement system can assist antibodies and phagocytes to kill pathogenic microorganisms in the immune activities. Additionally, it can act synergistically with GH and IGF-1 to improve the physiological and biochemical processes of animals, enhancing protein and fat metabolism in the liver, improving amino acid utilization, reducing body fat deposition, and increasing intermuscular fat. A high complement content can assist antibodies and phagocytes to kill pathogenic microorganisms [44]. This study found that the C4 content in GC300 was higher than that in the control group on the 10th day of lactation, and the effect was not significant on the 1st and 21st day of lactation, indicating that adding appropriate SI and APS mixture to the diet had a minor or no effect on the production of complement by lactating sows.

### 4.5. Effects of SI and APS Mixture on Serum Hormone Level of Lactating Sows

The adenohypophysis continuous secretion of PRL is necessary to maintain lactation [45]. PRL can increase the estrogen receptor content in mammary tissue, which stimulates the mammary epithelial cells to milk [46]. Furthermore, PRL can cooperate with GH to regulate substance metabolism and reasonably distribute nutrients to different tissues, which ensures that the mammary cells can absorb rich nutrients to form milk [47]. GH and PRL also involve in the synthesis of thymic epithelial cells and secretion of thymosin, which indirectly regulates immune function through thymosin. Our study showed that the PRL contents in GC200 and GC300 were higher than those in GC on the 1st and 10th day of lactation, and on the 21st day of lactation in GC300 PRL was higher compared with GC. The GH content in the GC200 was significantly higher than that in GC during the experimental period. The most significant effect on PRL and GH contents was observed in the GC200 and/or GC300. Soy flavone and genistein could significantly increase the PRL and GH levels in the serum of Holstein cows in the middle stage of lactation [48]. Additionally, Li [24] found that SI significantly increased the GH and PRL levels in the blood of lactating sows. Furthermore, the higher lactation yield in GC200 compared to GC also reflected that the SI and APS mixture improved the PRL and GH contents in the serum of lactating sows, and there may be a dose-effect as it was not higher in GC300.

IGF-1 is mainly found in the blood of animals, and most IGF-1 in serum comes from the liver [47]. In animals with anorexia nervosa, hunger, and malnutrition, GH content increases, while IGF-1 content decreases. IGF-1 levels in the blood can be used to measure the general health of animals [49]. Our study showed that IGF-1 contents in lactating sows were increased in GC200 and GC300 compared with those in GC. Thus, the appropriate amount of SI and APS mixture can improve the IGF-1 content in lactating sows and maintain the normal nutritional level and health status of animals.

During delivery and lactation, the physiological mechanism and nutritional metabolism of mammals would undergo drastic changes, especially the metabolism of the endocrine system, fat and protein metabolism, and a large amount of energy reacts with oxygen, accompanied by the production of free radicals [50,51]. T-AOC can directly evaluate antioxidant enzyme activity and non-enzymatic antioxidant capacity in animals, which reflects the strength of the defense system’s antioxidant capacity and is closely related to health [47,52]. In this experiment, the T-AOC content in GC200 was higher than that in GC on the 1st and 21st day of lactation, showing that the SI and APS mixture could strengthen the defense system against oxidation in lactating sows.

In the animal body, the antioxidant system can be divided into non-enzyme and enzyme systems according to the different factors involved in the antioxidant process. The antioxidant enzymes in the enzyme system include SOD, GSH-PX, and CAT. When the free radicals generated during the body’s peroxidation attack polyunsaturated fatty acids, a lipid peroxidation reaction is triggered, forming the final lipid peroxide MDA, which further leads to a certain degree of damage to the animal cells. The overall antioxidant capacity can be determined by detecting SOD, GSH-Px, CAT, and MDA contents. In this study, SI and APS mixture could increase SOD and GSH-PX contents in GC200 and GC300 on the 10th and 21st day of lactation, and increased CAT content in GC200 on the 21st day of lactation, and decreased MDA content in GC200 during the experimental period compared with those in GC. Wang [37] found that APS could increase SOD and GSH-PX contents and reduced MDA content in serum. Additionally, Hu [15] reported that the SI improved the plasma T-AOC and CAT activities, reduced MDA content, and improved the antioxidant capacity of lactating sows, which is consistent with the results of this experiment.

## 5. Conclusions

The effects of SI and APS mixture on the colostrum components, and serum antioxidant, immune, and hormone levels of lactating sows in the different treatment groups were assessed, and these parameters were compared with those in control. At an overall level, the SI and APS mixture can improve the lactation yield, serum immune ability, antioxidant performance and hormone levels of lactating sows, as well as their nutritional health status. The optimum dosage is 200 mg/kg.

## Figures and Tables

**Table 1 animals-11-00132-t001:** Composition (kg/100 kg) of the basal experimental diets ^1^ for gestating and lactating sows.

Items	Content
Ingredients	
Corn	69.00
Wheat bran	3.00
Soybean meal	19.00
Fish meal	2.00
Soybean oil	2.60
Calcium hydrogen phosphate	0.70
Limestone	1.00
Salt	0.70
Premix ^2^	2.00
Total, kg	100.00
Nutrient levels, on air-dry basis:	
Digestible energy ^3^, DE, MJ/kg	13.98
Crude protein ^4^, CP, %	16.35
Calcium ^4^, Ca, %	0.73
Phosphorus ^4^, P, %	0.34
Lysine ^4^, Lys, %	0.92
Methionine ^4^, Met, %	0.26
Threonine ^4^, Thr, %	0.59

^1^ Based on the NRC (1998; 2012) nutrient requirements for lactating sows. ^2^ The premix provided the following per kg of diet: VA 2000 IU, VD 200 IU, VE 45 IU, VK 0.5 mg, VB_1_ 1 mg, pantothenic acid 12 mg, nicotinic acid 10.25 mg, VB_6_ 3.85 mg, VB_12_ 15 ug, folic acid 1.35 mg, biotin 0.21 mg, VC 200 mg, Mn as manganese sulfate 20 mg, Fe as ferrous sulfate 80 mg, Cu as copper sulfate 5 mg, I as potassium iodide 0.14 mg, Se as sodium selenite 0.15 mg. ^3^ Calculated value (NRC, 1998; 2012). ^4^ Analysed content.

**Table 2 animals-11-00132-t002:** Effects of SI and APS mixture on the production performance of lactating sows.

Items	GC	GC100	GC200	GC300
ADFI ^1^, kg/d	6.39 ± 0.19 ^b^	6.84 ± 0.43 ^a^	7.05 ± 0.51 ^a^	6.54 ± 0.51 ^b^
ADG ^2^, g/d	217 ± 28	248 ± 13	260 ± 4	222 ± 8
Total lactation yield, kg	147.4 ± 6.4 ^b^	168.0 ± 8.9 ^ab^	175.6 ± 5.4 ^a^	146.0 ± 8.2 ^b^
Diarrhea ratio of piglets, %	7.26 ± 2.28	6.87 ± 2.26	6.11 ± 2.90	6.49 ± 2.20
Mortality ratio of piglets, %	12.7 ± 2.2	9.1 ± 3.2	9.1 ± 3.2	10.9 ± 2.7

^1^ Average daily feed intake of lactating sows.^2^ Average daily gain of piglets. ^ab^ In the same row, values with the same small or no letter superscripts mean no significant difference (*p* > 0.05), and with different small letter superscripts mean significant difference (*p* < 0.05).

**Table 3 animals-11-00132-t003:** Effects of SI and APS mixture on colostrum composition of lactating sows.

Items	GC	GC100	GC200	GC300
Fat content, %	5.27 ± 0.41	5.36 ± 0.16	5.40 ± 0.10	5.37 ± 0.31
Lactose content, %	5.36 ± 0.12	5.33 ± 0.20	5.31 ± 0.27	5.30 ± 0.28
protein content, %	8.64 ± 0.30	8.66 ± 0.18	8.66 ± 0.40	8.65 ± 0.31

In the same row, values with the same small or no letter superscripts mean no significant difference (*p* > 0.05), and with different small letter superscripts mean significant difference (*p* < 0.05).

**Table 4 animals-11-00132-t004:** Effects of SI and APS mixture on serum antioxidant and other indexes of lactating sows.

Items	GC	GC100	GC200	GC300
1st day of lactation				
TP, g/L	63.54 ± 0.52	65.72 ± 0.45	66.68 ± 0.46	64.94 ± 2.01
ALB, g/L	27.18 ± 0.67 ^b^	30.35 ± 0.77 ^ab^	32.90 ± 0.48 ^a^	32.59 ± 0.93 ^a^
UN, mmol/L	7.05 ± 0.05 ^a^	5.14 ± 0.09 ^c^	4.86 ± 0.08 ^b^	6.12 ± 0.25 ^c^
G, mmol/L	3.97 ± 0.20	3.86 ± 0.11	3.88 ± 0.07	3.51 ± 0.08
TG, mmol/L	3.30 ± 0.03 ^a^	2.86 ± 0.02 ^b^	2.65 ± 0.05 ^b^	2.78 ± 0.06 ^b^
CHO, mmol/L	2.30 ± 0.04 ^b^	1.99 ± 0.10 ^a^	1.97 ± 0.03 ^a^	2.02 ± 0.09 ^a^
10th day of lactation				
TP, g/L	53.61 ± 1.92 ^b^	56.67 ± 2.02 ^ab^	65.54 ± 0.64 ^a^	62.00 ± 1.46 ^ab^
ALB, g/L	27.86 ± 0.81	28.48 ± 0.41	30.02 ± 0.19	28.76 ± 0.60
UN, mmol/L	8.04 ± 0.07 ^a^	5.52 ± 0.38 ^b^	5.36 ± 0.10 ^b^	6.89 ± 0.35 ^ab^
G, mmol/L	4.23 ± 0.22	3.91 ± 0.04	3.61 ± 0.07	3.77 ± 0.11
TG, mmol/L	3.21 ± 0.02 ^a^	2.99 ± 0.06 ^b^	2.89 ± 0.05 ^b^	2.83 ± 0.06 ^b^
CHO, mmol/L	2.87 ± 0.07 ^b^	2.48 ± 0.03 ^a^	2.36 ± 0.07 ^a^	2.42 ± 0.03 ^a^
21st day of lactation				
TP, g/L	54.24 ± 1.25 ^b^	56.59 ± 1.07 ^ab^	62.19 ± 1.13 ^a^	60.85 ± 1.23 ^ab^
ALB, g/L	25.77 ± 0.77	27.23 ± 0.84	29.20 ± 0.44	29.59 ± 0.92
UN, mmol/L	6.77 ± 0.51 ^a^	5.83 ± 0.28 ^b^	5.52 ± 0.08 ^b^	5.65 ± 0.16 ^b^
G, mmol/L	3.89 ± 0.21	3.70 ± 0.07	3.41 ± 0.17	3.42 ± 0.07
TG, mmol/L	2.90 ± 0.04 ^a^	2.53 ± 0.02 ^b^	2.51 ± 0.08 ^b^	2.92 ± 0.03 ^a^
CHO, mmol/L	2.24 ± 0.07 ^b^	1.88 ± 0.10 ^a^	1.83 ± 0.03 ^a^	1.97 ± 0.05 ^a^

^ab^ In the same row, values with the same small or no letter superscripts mean no significant difference (*p* > 0.05), and with different small letter superscripts mean significant difference (*p* < 0.05). TP: Total protein; ALB: Albumin; UN: Urea nitrogen; G: Glucose; TG: Triglyceride; CHO: Total cholesterol.

**Table 5 animals-11-00132-t005:** Effects of SI and APS mixture on serum immune indexes of lactating sows.

Items	GC	GC100	GC200	GC300
1st day of lactation				
IL-2, ng/mL	0.28 ± 0.01	0.32 ± 0.01	0.30 ± 0.01	0.28 ± 0.01
TNF-α, ng/m/L	0.35 ± 0.01 ^b^	0.36 ± 0.01 ^ab^	0.38 ± 0.01 ^a^	0.36 ± 0.01 ^ab^
C4, μg/mL	62.68 ± 1.23	63.86 ± 0.69	65.85 ± 0.20	63.62 ± 0.74
IgG, μg/mL	429.23 ± 15.21	435.52 ± 6.85	454.42 ± 4.80	455.19 ± 4.35
IgA, μg/mL	71.01 ± 0.67 ^b^	72.27 ± 0.57 ^b^	77.50 ± 0.61 ^a^	75.62 ± 1.16 ^a^
10th day of lactation				
IL-2, ng/mL	0.27 ± 0.01 ^b^	0.32 ± 0.01 ^a^	0.35 ± 0.01 ^a^	0.35 ± 0.01 ^a^
TNF-α, ng/m/L	0.27 ± 0.01 ^b^	0.28 ± 0.01 ^b^	0.31 ± 0.01 ^a^	0.30 ± 0.01 ^ab^
C4, μg/mL	66.80 ± 0.92 ^b^	68.04 ± 0.58 ^ab^	70.57 ± 0.83 ^ab^	71.69 ± 0.37 ^a^
IgG, μg/mL	408.35 ± 12.87 ^b^	445.93 ± 13.85 ^ab^	479.64 ± 3.27 ^a^	460.90 ± 7.78 ^ab^
IgA, μg/mL	74.28 ± 0.32^b^	74.73 ± 1.36 ^b^	81.41 ± 0.56 ^a^	76.26 ± 0.72 ^b^
21st day of lactation				
IL-2, ng/mL	0.26 ± 0.002 ^c^	0.27 ± 0.01 ^c^	0.32 ± 0.01 ^a^	0.29 ± 0.01 ^b^
TNF-α, ng/m/L	0.33 ± 0.01 ^b^	0.35 ± 0.01 ^ab^	0.38 ± 0.01 ^a^	0.36 ± 0.01 ^ab^
C4, μg/mL	63.90 ± 1.01	65.12 ± 0.94	65.02 ± 1.39	66.71 ± 0.62
IgG, μg/mL	327.05 ± 2.62 ^c^	334.25 ± 6.08 ^bc^	393.67 ± 5.12 ^a^	366.97 ± 7.77 ^ab^
IgA, μg/mL	69.88 ± 0.99 ^b^	70.94 ± 2.59 ^b^	76.58 ± 1.32 ^a^	73.90 ± 0.40 ^b^

^abc^ In the same row, values with the same small or no letter superscripts mean no significant difference (*p* > 0.05), and with different small letter superscripts mean significant difference (*p* < 0.05). IL-2: Interleukin-2; TNF-α: Tumor Necrosis Factor-α; C4: Complement 4; IgG: Immunoglobulin G; IgA: Immunoglobulin A.

**Table 6 animals-11-00132-t006:** Effects of SI and APS mixture on serum hormone level of lactating sows.

Items	GC	GC100	GC200	GC300
1st d of lactation				
PRL, ng/mL	0.81 ± 0.01 ^b^	0.83 ± 0.01 ^ab^	0.84 ± 0.00 ^a^	0.84 ± 0.01 ^a^
GH, ng/mL	22.57 ± 0.37 ^b^	23.92 ± 0.11 ^ab^	24.56 ± 0.18 ^a^	23.69 ± 0.40 ^ab^
IGF-1, ng/mL	194.20 ± 5.69 ^b^	206.07 ± 3.92 ^b^	239.20 ± 3.53 ^a^	213.76 ± 2.33 ^a^
T-AOC, U/mL	7.62 ± 0.02 ^b^	7.67 ± 0.02 ^ab^	7.77 ± 0.04 ^a^	7.66 ± 0.02 ^ab^
SOD, μg/mL	0.33 ± 0.01 ^c^	0.35 ± 0.01 ^bc^	0.37 ± 0.01 ^ab^	0.39 ± 0.01 ^a^
GSH-Px, pg/mL	63.30 ± 0.24 ^b^	64.19 ± 0.19 ^a^	64.71 ± 0.28 ^a^	65.14 ± 0.27 ^a^
CAT, ng/mL	17.33 ± 0.49	17.75 ± 0.038	18.23 ± 0.17	18.49 ± 0.50
MDA, mmol/mL	6.77 ± 0.03 ^a^	6.42 ± 0.16 ^a^	5.69 ± 0.05 ^b^	5.61 ± 0.06 ^b^
10th d of lactation				
PRL, ng/mL	0.72 ± 0.01 ^b^	0.74 ± 0.06 ^ab^	0.77 ± 0.02 ^a^	0.77 ± 0.02 ^a^
GH, ng/mL	24.04 ± 0.35 ^c^	25.02 ± 0.49 ^b^	28.16 ± 0.54 ^a^	27.01 ± 0.30 ^ab^
IGF-1, ng/mL	164.15 ± 2.47 ^c^	182.38 ± 3.13 ^bc^	203.87 ± 3.88 ^a^	196.70 ± 2.71 ^ab^
T-AOC, U/mL	7.73 ± 0.02	7.83 ± 0.06	7.90 ± 0.06	7.95 ± 0.02
SOD, μg/mL	0.34 ± 0.01 ^b^	0.38 ± 0.01 ^a^	0.41 ± 0.01 ^a^	0.38 ± 0.01 ^a^
GSH-Px, pg/mL	81.36 ± 0.24 ^b^	82.43 ± 0.45 ^ab^	83.69 ± 0.17 ^a^	82.13 ± 0.22 ^ab^
CAT, ng/mL	18.98 ± 0.17	19.20 ± 0.15	20.62 ± 0.33	20.39 ± 0.32
MDA, mmol/mL	7.78 ± 0.03 ^a^	7.01 ± 0.03 ^b^	5.37 ± 0.06 ^c^	5.58 ± 0.07 ^c^
21st d of lactation				
PRL, ng/mL	0.64 ± 0.01 ^c^	0.65 ± 0.01 ^bc^	0.67 ± 0.01 ^ab^	0.68 ± 0.01 ^a^
GH, ng/mL	23.82 ± 0.26 ^c^	25.52 ± 0.31 ^bc^	27.89 ± 0.45 ^a^	26.51 ± 0.46 ^ab^
IGF-1, ng/mL	209.78 ± 4.10 ^c^	217.92 ± 5.54 ^bc^	253.47 ± 3.36 ^a^	240.89 ± 2.20 ^ab^
T-AOC, U/mL	7.76 ± 0.06 ^b^	8.03 ± 0.05 ^ab^	8.13 ± 0.06 ^a^	8.08 ± 0.03 ^a^
SOD, μg/mL	0.33 ± 0.01 ^b^	0.40 ± 0.002 ^a^	0.41 ± 0.01 ^a^	0.38 ± 0.01 ^a^
GSH-Px, pg/mL	67.35 ± 0.55 ^b^	70.66 ± 0.35 ^a^	72.47 ± 0.33 ^a^	72.82 ± 0.83 ^a^
CAT, ng/mL	17.71 ± 0.42 ^b^	20.04 ± 0.39 ^ab^	20.29 ± 0.35 ^a^	18.52 ± 0.46 ^ab^
MDA, mmol/mL	6.80 ± 0.08 ^a^	6.59 ± 0.09 ^a^	5.66 ± 0.05 ^b^	6.60 ± 0.08 ^a^

^abc^ In the same row, values with the same small or no letter superscripts mean no significant difference (*p* > 0.05), and with different small letter superscripts mean significant difference (*p* < 0.05). PRL: Prolactin; GH: Growth hormone; IGF-1: Insulin-like growth factor-1; T-AOC: Total antioxidant capacity, SOD: Activities of superoxide dismutase; GSH-Px: Glutathione peroxidase; CAT: Catalase; MDA: Malondialdehyde.

## Data Availability

Data presented are original and not inappropriately selected, manipulated, enhanced, or fabricated.

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
