# Peer review of "Effects of Soybean Isoflavone and Astragalus Polysaccharide Mixture on Colostrum Components, Serum Antioxidant, Immune and Hormone Levels of Lactating Sows"

_animals, 2021, doi:10.3390/ani11010132_

Round 1
Reviewer 1 Report
Despite the modifications made by the authors that improved the manuscript I still think that the document has too many serious flaws. I attach a document with my questions, corrections and suggestions but at this stage I have to recommend its rejection. My main concerns are related to:
- the statement of differences or effects (both in Results and in Discussion) when statistically there are no differences between treatments. Numerical differences cannot be used to claim possible effects of the experimental diets.
- results section is still very confusing with a lot of text, mixed with differences in %, highlighting many times values that were not different, completely losing the focus
- I still find some statistical differences “strange” (e.g. TNF-α on the 1st day of lactation) while I wonder if there no some differences at a lower P-value than 0.05 (all differences in the manuscript are for P<0.05….)
- the discussion section was improved but I still think it is not clear to read and understand and mostly I think it has various wrong references (papers that are not connected to the described subject), e.g. lines 349, 411, 431
- the references list has a completely diverse citation format

Author Response
Dear Reviewer:
Thank you for comments concerning our manuscript entitled Effects of soybean isoflavone and astragalus polysaccharide mixture on colostrum components, serum antioxidant, immune and hormone levels of lactating sows, ID: animals-102987. Those comments are all valuable and very helpful for revising and improving our paper, as well as the important guiding significance to our researcher. We have studied comments carefully and have made a correction which we hope meet with approval. Revised portions are marked in red on the paper. The main correction in the paper and the responses to the editor and reviewer’s comments are as flowing:
Responds to the reviewer’s comments:
Reviewer #1:
Comments: Despite the modifications made by the authors that improved the manuscript I still think that the document has too many serious flaws. I attach a document with my questions, corrections and suggestions but at this stage I have to recommend its rejection.
Response: Thank you for your valuable comments and your meticulous revisions in the document. We have made modifications according to your requirements in the attached paper, and we hope to meet your requirements.
Concern #1: the statement of differences or effects (both in Results and in Discussion) when statistically there are no differences between treatments. Numerical differences cannot be used to claim possible effects of the experimental diets.
Response: Thank you. We have re-written all the result parts and the corresponding parts in discussion according to the reviewer’s suggestion.
Concern #2: results section is still very confusing with a lot of text, mixed with differences in %, highlighting many times values that were not different, completely losing the focus
Response: Thank you. We have re-written all the result parts and deleted the comparisons with no significant differences according to the reviewer’s suggestion.
Concern #3: I still find some statistical differences “strange” (e.g. TNF-α on the 1st day of lactation) while I wonder if there no some differences at a lower P-value than 0.05 (all differences in the manuscript are for P<0.05….)
Response: Thank you. We have re-written the result parts according to the reviewer’s suggestion.
Concern #4: the discussion section was improved but I still think it is not clear to read and understand and mostly I think it has various wrong references (papers that are not connected to the described subject), e.g. lines 349, 411, 431
Response: Thank you. We have replaced the inappropriate references in accordance with the reviewer’s suggestion and made corresponding modifications in the discussion section.
Concern #5: the references list has a completely diverse citation format
Response: Thank you. We have modified the format of the references according to Animals’s requirements
Special thanks to you for your good comments.
Asso. Prof. Xinping Diao, E-mail: diaoxp63@163.com
Prof. Li Xu, E-mail: xuli_19621991@163.com

Reviewer 2 Report
Comment on statistics: be careful that when measures are repeated on a given animal, the interanimal variance must be added to the residual one, due to autoregressive covariance structure. This always reduces the significiativity of the results. So, probably some results could be erroneously significant.
Results: the extensive presentation of the results, as percentage of increases, could be considered as a bit excessive, since they could be deduced from the tables.
Discussion: the discussion provides a good overview on animal physiology, although some considerations could be regarded as well known (f.e. L332-335, L408-411). On the same way, L 365-375: the authors discuss about kidney diseases, while the experimental sows were assumed healthy. Plasma indicators of kidney malfunction are visible at an advanced state of disease.
Physiologicaly, the plasma GH concentration is highly variable, owing to its pulsatile secretion nature. It would be worth to mention it.
4.5.: refer to "antioxydant activity" in the title
Form:
Line 214: "difference" instead of "different"
Line 215: a space is lacking
Line 243, 249, 271: misplaced points
Lin 370: "urine nitrogen" rather than "unrine protein"
Line 421: "complement" instead of "compliment"
Read carefully the whole paper.
Author Response
Dear Reviewer:
Thank you for comments concerning our manuscript entitled Effects of soybean isoflavone and astragalus polysaccharide mixture on colostrum components, serum antioxidant, immune and hormone levels of lactating sows, ID: animals-102987. Those comments are all valuable and very helpful for revising and improving our paper, as well as the important guiding significance to our researcher. We have studied comments carefully and have made a correction which we hope meet with approval. Revised portions are marked in red on the paper. The main correction in the paper and the responses to the editor and reviewer’s comments are as flowing:
Responds to the reviewer’s comments:
Reviewer #2:
Comments:
Comment on statistics: be careful that when measures are repeated on a given animal, the interanimal variance must be added to the residual one, due to autoregressive covariance structure. This always reduces the significiativity of the results. So, probably some results could be erroneously significant.
Results: the extensive presentation of the results, as percentage of increases, could be considered as a bit excessive, since they could be deduced from the tables.
Discussion: the discussion provides a good overview on animal physiology, although some considerations could be regarded as well known (f.e. L332-335, L408-411). On the same way, L 365-375: the authors discuss about kidney diseases, while the experimental sows were assumed healthy. Plasma indicators of kidney malfunction are visible at an advanced state of disease.
Physiologicaly, the plasma GH concentration is highly variable, owing to its pulsatile secretion nature. It would be worth to mention it.
Response: Thank you for your valuable comments. The colostrum composition and serum antioxidant, immune, and hormone levels of each sow were calculated, and the results were expressed as mean ± SEM.
In the modern formal pig farm, a healthy sow is eliminated after six parities, so there is little difference in biochemical parameters between the gestating and lactating sows before the 6th parity. In this experiment, the sows, with same parity (second parity) were selected to reduce the influencing factors of the experimental results to make the results more accurate. We have readjusted the results section and deleted the percentage form of the results section. We have revised the discussion according to the reviewer’s suggestion.
Concern #1: 4.5.: refer to "antioxydant activity" in the title
Response: Thank you. We have modified it as Effects of soybean isoflavone and astragalus polysaccharide mixture on colostrum components, serum antioxidant, immune and hormone levels of lactating sows according to the reviewer’s advice
Concern #2: Line 214: "difference" instead of "different"
Response: Thank you. We have modified it according to the reviewer’s advice.
Concern #3: Line 215: a space is lacking
Response: Thank you. We have modified it according to the reviewer’s advice.
Concern #4: Line 243, 249, 271: misplaced points
Response: Thank you. We have adjusted them according to the reviewer’s advice.
Concern #5: Lin 370: "urine nitrogen" rather than "unrine protein"
Response: Thank you. We have modified it according to the reviewer’s advice.
Concern #6: Line 421: "complement" instead of "compliment"
Response: Thank you. We have modified it according to the reviewer’s advice.
Concern #7: Read carefully the whole paper.
Response: Thank you. We will read the paper carefully.
Special thanks to you for your good comments.
Asso. Prof. Xinping Diao, E-mail: diaoxp63@163.com
Prof. Li Xu, E-mail: xuli_19621991@163.com

Reviewer 3 Report
GENERAL COMMENTS
The authors report the results of an articulated trial regarding several aspects of the administration of an isoflavone (soybean) and a polysaccharide (Astragalus) to lactating sows.
The research is, in my opinion, well conducted and quite complex and complete. I think it worths te publication on Animals.
Notwithstanding, I have some observations regarding the manuscript, reported in the following section.
SPECIFIC COMMENTS
- Some English errors and several typos are present
- The main concern is about the statistical analysis of data: it s a pity that the time of colostrum/milk collection had not be considered, since it could express the variability of these interesting variables during time. Moreover, taking into account the time of collection, the variability dur to collection time could be excluded from the total variability for the single variables, exposing the results to a more defined treatment effect.
- Maybe, the adoption of a two-way ANOVA (not repeated measures one) could help.
Author Response
Dear Reviewer:
Thank you for comments concerning our manuscript entitled Effects of soybean isoflavone and astragalus polysaccharide mixture on colostrum components, serum antioxidant, immune and hormone levels of lactating sows, ID: animals-102987. Those comments are all valuable and very helpful for revising and improving our paper, as well as the important guiding significance to our researcher. We have studied comments carefully and have made a correction which we hope meet with approval. Revised portions are marked in red on the paper. The main correction in the paper and the responses to the editor and reviewer’s comments are as flowing:
Responds to the reviewer’s comments:
Reviewer #3:
Comments:
GENERAL COMMENTS
The authors report the results of an articulated trial regarding several aspects of the administration of an isoflavone (soybean) and a polysaccharide (Astragalus) to lactating sows.
The research is, in my opinion, well conducted and quite complex and complete. I think it worths te publication on Animals.
Notwithstanding, I have some observations regarding the manuscript, reported in the following section.
Response: Thank you for your recognition of our work.
SPECIFIC COMMENTS
Some English errors and several typos are present
The main concern is about the statistical analysis of data: it s a pity that the time of colostrum/milk collection had not be considered, since it could express the variability of these interesting variables during time. Moreover, taking into account the time of collection, the variability dur to collection time could be excluded from the total variability for the single variables, exposing the results to a more defined treatment effect.
Maybe, the adoption of a two-way ANOVA (not repeated measures one) could help.
Response: Thank you. We have carefully corrected the writing errors in this article. The pig farm has a complete set of management measures for sows, and the entire farrowing process of sows will end in 30 minutes, so we thought that the time of collecting colostrum has little or no effect on the test results of the samples. We consulted the two corresponding authors, who said that our experimental design was a simple single-factor design to verify the effects of SI and APS on lactation sows. ANOVA and Tukey’s honestly significant difference pos-thoc test used for statistical comparisons of different treatments in this article can be one of the possible statistical ways. Thank you very much.
Special thanks to you for your good comments.
Asso. Prof. Xinping Diao, E-mail: diaoxp63@163.com
Prof. Li Xu, E-mail: xuli_19621991@163.com

Round 2
Reviewer 1 Report
The manuscript is improved but it still have some things to improve and correct. You have to explain how was milk yield calculated, you have to better explain some of the effects found or not found by you when compared to the literature and check and correct some references that aren't good or even incorrect. All these critics and suggestions and other are in the revised manuscript. The tables must be improved, please check.

Author Response
Dear Reviewer:
Thank you for comments concerning our manuscript entitled Effects of soybean isoflavone and astragalus polysaccharide mixture on colostrum components, serum antioxidant, immune and hormone levels of lactating sows, ID: animals-1029781. Those comments are all valuable and very helpful for revising and improving our paper, as well as the important guiding significance to our researcher. We have studied comments carefully and have made a correction which we hope meet with approval. Revised portions are marked in red on the paper. The main correction in the paper and the responses to the editor and reviewer’s comments are as flowing:
Responds to the reviewer’s comments:
Comments: The manuscript is improved but it still have some things to improve and correct. You have to explain how was milk yield calculated, you have to better explain some of the effects found or not found by you when compared to the literature and check and correct some references that aren't good or even incorrect. All these critics and suggestions and other are in the revised manuscript. The tables must be improved, please check.
Response: Thank you for your valuable comments and your meticulous revisions in the document. We have made modifications according to your requirements in the attached paper, and we hope to meet your requirements. We have replaced the inappropriate references and made corresponding modifications in the manuscript.
Special thanks to you for your good comments.
Asso. Prof. Xinping Diao, E-mail: diaoxp63@163.com
Prof. Li Xu, E-mail: xuli_19621991@163.com

This manuscript is a resubmission of an earlier submission. The following is a list of the peer review reports and author responses from that submission.
Round 1
Reviewer 1 Report
Comments on manuscript Animals-987947 “Effects of soybean isoflavone and astragalus polysaccharide mixture on colostrum components, serum biochemical, immune and hormone level of lactating sows”
The research topic is interesting and with potential use in an extremely important and challenging phase of the sow reproductive life, the transition from gestation to lactation.
I’m not a native English speaker but I consider the English language and style of the manuscript very poor. It makes extremely hard to follow and understand all manuscript.
For clarity, the simple summary shouldn’t have the exhaustive list of the analyzed compounds.
As in the Abstract, it should indicate that studied mixture was added to the sow’s diet (and in the M&M section how was it added/mixed).
Authors say that the study was approved by Animal Care and Use Committee of Northeast Agricultural University but there is no mention to any process number or ID.
Line 71: what authors mean with “with similar genetic background”?
The M&M section lacks of several information:
- the sow’s parity (mean and extremes)
- how was the mixture added to the diets (as mention above) and in the gestating phase how was controlled the ingestion? Were they in individual stalls?
- the basal diet was equal during gestation and lactation phases?
- farrowing information (e.g. total born, born alive, stillbirths) here or in results section. Litter size (and balance) after cross-fostering?
- What means “The other feeding management and immunization procedures were carried out following the pig farm” (lines 91-92)?
- lines 93-94 – 10 ml of colostrum (not milk). “2h after delivery”? 2h after farrowing start or end? If end, how many hours of difference could exist between samples (between sows)?
- the specific kits reference and the inter and intra assay variation should be indicated
Besides the above mention information and yet in the M&M section:
Line 85-86 – poor English and information about piglet should be latter (after farrowing information)
Line 87- 3 days…before the previewed farrowing date, isn’t it?
Line 90 – was do you mean with “close to ad libitum”?
Lines 109-110 – why not fat, lactose and protein contents in colostrum? The analyzer information is repeated
Finally, and for clarity mainly in the results and discussion section I suggest alternative designations of the groups (after their correct definition). For example: GC, G100, G200, G300
Results section:
- the manuscript should have results on piglet’s survival and growth, why weren’t they presented? Even to confirm some potential benefits for the nursing piglets….
- Lines 132-133: why the authors highlight the very small differences between treatments. They are not statistically significant and even if they were, what is the biological relevance of the ingestion of colostrum with 5.36% or 5.40% of fat??? The difference between the control diet group with the others should be checked (with a SEM of 0.4 in the control group it is not obvious to have significant differences to the other groups)
- All results texts are too confusing and exhaustive, they should start saying the day of sampling at first and they should be reduced, many times they just repeat the tables information
- Assuming that there aren’t comparisons within groups for sampling time (and I ask why those analyzes weren’t made) I ask why table 5 has a different format than the previous ones?
Discussion
The discussion section lacks of clarity (mostly because of the poor English). When literature is cited with the author name perhaps the numeric reference should be placed right after that citation. I don’t think “some scholars” (liens 280-281 and 321) is an appropriate way to cite. I also would prefer the use of “mammary” instead of “breast” (used several times in the discussion) and “farrowing” instead of “delivery”.
Lines 274-275 – if there is a hypoglycemic effect why there were no significant differences in serum glucose of the sows?
Line 315 – the word “genistein” is used for the first time and it should explained/defined
In lines 329, 339, 376 the authors claim significant and permanent (during the time) differences that aren’t in accordance with the presented results, therefore those sentences can’t be written in scientific paper
Looking at the authors contributions, there are 2 authors that wrote the manuscript while the others just read and approve it, is that a really significant contribution? No mentions are made to the experiment supervision, sampling, laboratory work….
Reviewer 2 Report
General:
English (spelling, grammar,...) use must be completely revised.
Specific:
L 17-24: avoid the list as such. Give an overall consideration.
L32: remove "Results:"
Statistics:
- for repeated measurements, mixed model with autoregressive covariance structure is required.
- provide also the overall significances of the models.
- perform and discuss orthogonal contrasts (linear and quadratic) rather than extensive descriptions group by group.
- do not describe or discuss what is not significant or irrelevant
Tables:
Limit decimal number: more than 3 significant figures are genetaly useless
Give meanings of the abbreviations
Results
As such, it is very difficult to make an opinion about the presented effets. Reconsider the statistical approach (see above).
L90: give details on how piglets number adjustment was performed.
L131-132: such increases are irrelevant.
Discussion
L258-264 and elsewhere: avoid generalities. Discuss your results and then support them with actual knowledges.
Avoid to repeat exceedingly some description already made in the "results" section.